# RNAS-CL: Robust Neural Architecture Search by Cross-Layer Knowledge Distillation

## Abstract

Deep Neural Networks are vulnerable to adversarial attacks. Neural Architecture Search (NAS), one of the driving tools of deep neural networks, demonstrates superior performance in prediction accuracy in various machine learning applications. However, it is unclear how it performs against adversarial attacks. Given the presence of a robust teacher, it would be interesting to investigate if NAS would produce robust neural architecture by inheriting robustness from the teacher. In this paper, we propose Robust Neural Architecture Search by Cross-Layer Knowledge Distillation (RNAS-CL), a novel NAS algorithm that improves the robustness of NAS by learning from a robust teacher through cross-layer knowledge distillation. Unlike previous knowledge distillation methods that encourage close student/teacher output only in the last layer, RNAS-CL automatically searches for the best teacher layer to supervise each student layer. Experimental result evidences the effectiveness of RNAS-CL and shows that RNAS-CL produces small and robust neural architecture. The code of RNAS-CL is available at https://anonymous.4open.science/r/RNAS-CL-06A0/.

## 1 Introduction

Neural Architecture Search (NAS), one of the most promising driving tools with state-of-the-art performance of deep neural networks in various tasks such as computer vision and natural language processing, has been attracting a lot of attention in recent years. NAS automatically searches for neural architecture according to user-specified criteria without human intervention, thus avoiding the time-consuming and burdensome manual design of neural architecture. Earlier studies in NAS are based on Evolutionary Algorithms (EA) (Real et al., 2017) and Reinforcement Learning (RL) (Zoph & Le, 2017; Tan et al., 2019). However, despite their performance, they are computationally expensive. It would take them more than 3000 GPU days to achieve state-of-the-art performance on the ImageNet dataset. Most recent studies (Liu et al., 2019; Cai et al., 2019; Wu et al., 2019; Wan et al., 2020; Nath et al., 2020) encode architectures as a weight-sharing super-net and optimize the weights using gradient descent. Architectures found by NAS exhibit two significant advantages. First, they achieve SOTA performance for various computer vision tasks. Second, the architectures found by NAS are efficient in terms of speed and size. Both advantages make NAS incredibly useful for real-world applications. However, most NAS methods are designed to optimize accuracy, parameters, or FLOPs. It is not clear how these architectures perform against adversarial attacks. In this paper, we propose RNAS-CL, a NAS method that jointly optimizes accuracy, latency, and robustness against adversarial attacks without robust training.

Adversarial attacks are performed by adding adversarial samples, for example, adding small sophisticated perturbations to the clean image, such that the model misclassifies the image. It is widely accepted that deep learning models are susceptible to adversarial attacks (Szegedy et al., 2014). Therefore, it is critical to analyze the robustness of models against adversarial attacks. Adversarial robust models are crucial for security-sensitive applications such as self-driving cars, health care, and surveillance cameras. For example, a self-driving car might not recognize a signboard after attaching a patch; in a surveillance system, an unauthorized person might get access by fooling the DNN model.

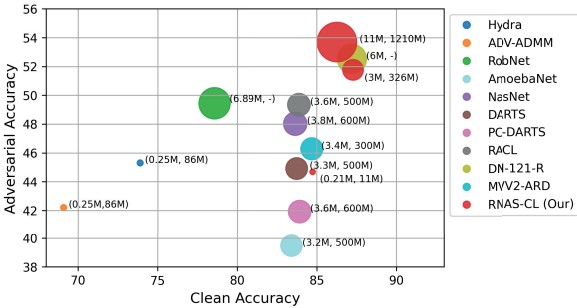

Figure 1: The figure compares various SOTA efficient and robust methods on CIFAR-10. Clean Accuracy represents top-1 accuracy on clean images. Adversarial Accuracy represents top-1 accuracy on images perturbed by PGD attack. A larger marker size indicates larger architecture. The numbers in brackets represent the number of parameters and MACs, respectively.

Adversarial training (Goodfellow et al., 2015; Madry et al., 2018; Kannan et al., 2018; Tramèr et al., 2018; Zhang et al., 2019a) is the most standard defense mechanism against adversarial attacks. Here, the models are trained on adversarial examples, which are often generated by fast gradient sign method (FGSM) (Goodfellow et al., 2015) or projected gradient descent (PGD) (Madry et al., 2018). Other types of defense mechanisms include models trained by losses or regularizations (Cissé et al., 2017; Hein & Andriushchenko, 2017; Yan et al., 2018; Pang et al., 2020), transforming inputs before feeding to model (Dziugaite et al., 2016; Guo et al., 2018; Xie et al., 2019), and using model ensemble (Kurakin et al., 2018; Liu et al., 2018).

Orthogonal to these methods, recent research (Madry et al., 2018; Guo et al., 2020; Su et al., 2018; Xie & Yuille, 2020; Huang et al., 2021) found an intrinsic influence of network architecture on adversarial robustness. Inspired by this idea, we propose Robust Knowledge Distillation for Neural Architecture Search (RNAS-CL), to the best of our knowledge, the first NAS method that uses knowledge distilled from a robust teacher model to find a robust architecture. Knowledge distillation transfers knowledge from a complex teacher model to a small student model. In standard knowledge distillation (Hinton et al., 2015), outputs from the teacher model are used as "soft labels" to train the student model. However, apart from the final teacher outputs, intermediate layers contain rich attention information. Different intermediate layers attend to different parts of the input object (Zagoruyko & Komodakis, 2017).

Hence, we ask the question: *can a robust teacher improve the robustness of the student model by providing information about where to look, i.e., where to pay attention?* The proposed RNAS-CL gives affirmative answers to the above question. In RNAS-CL, apart from learning from the output of the robust teacher model, each layer in the student learns "where to look" from the layers in the teacher model. However, the teacher and student might have a different number of layers. This leads us to another question regarding how to map a student layer to its corresponding teacher layer that it should learn from. In RNAS-CL, apart from searching the architecture of the student model, we search for the perfect tutor (teacher) layer for each student layer.

Let us consider a teacher ($T$) and student ($S$) model with $n_t$ and $n_s$ layers, respectively. $T_i, S_i$ are the $i$-th teacher and student layer, respectively. In RNAS-CL, each student layer $S_i$ is associated with $n_t$ gumbel weights, and each gumbel weight corresponds to each teacher layer. Intuitively, each gumbel weight indicates the weight of the connection between the student layer and each teacher layer. In the search phase, besides optimizing the architectural weights, we optimize these gumbel weights to find the perfect teacher layer. We hope the teacher to teach "where to pay attention." Therefore, by virtue of our RNAS-CL loss function for each student-teacher layer pair, each student layers learns robustness from a properly and automatically chosen teacher layer by maximizing the similarity of its attention map to that of its teacher layer.

## 1.1 CONTRIBUTIONS

Below are the main contributions of this work.

1. **Adversarial robust NAS.** RNAS-CL optimizes neural architecture to achieve a good tradeoff between robustness and prediction accuracy in a differentiable manner. To the best of our knowl-

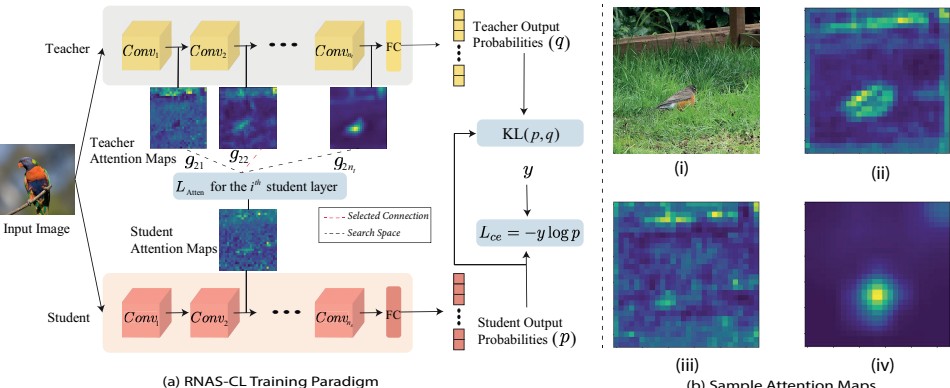

Figure 2: (a) Training paradigm based on RNAS-CL. We connect attention maps from each student layer to each robust teacher layer. For each student layer, we search for the optimum teacher layer. $g_{ij}$ represents gumbel weights associated between $i^{th}$ student layer and $j^{th}$ teacher layer. RNAS-CL induces robustness to the student model by searching for the optimum teacher layer. We also search for the number of filters in each layer to build an efficient model inspired by FBNetV2 (Wan et al., 2020). (b) Sample attention maps corresponding to input Image (i) from low-level (ii), mid-level (iii), and high-level (iv) convolution layers.

    edge, RNAS-CL is the first NAS method that optimizes robustness and prediction accuracy without robust training. Leveraging the penalty on model size/inference cost, the neural architecture found by RNAS-CL is compact compared to competing NAS methods. We compare RNAS-CL with other computationally efficient and robust models (Sehwag et al., 2020; Ye et al., 2019; Gui et al., 2019; Goldblum et al., 2020; Dong et al., 2020; Huang et al., 2021). Compared to these models, similar sized RNAS-CL models achieve up to $\sim 10\%$ higher clean accuracy and up to $\sim 5\%$ higher PGD accuracy on CIFAR-10 dataset.

2. **Cross-Layer Knowledge Distillation.** Our work advances the research of Knowledge Distillation (KD) using NAS. In particular, while conventional KD only uses fixed connections between teacher and student models to guide the student model, RNAS-CL extends the teaching scheme to learnable connections between layers of the teacher and the student models.

## 2 RELATED WORK

### 2.1 KNOWLEDGE DISTILLATION

Knowledge Distillation (KD) transfers knowledge from a large, cumbersome model to a small model. (Hinton et al., 2015) proposes the teacher-student model, where they use the soft targets from the teacher to train the student model. KD forces the student to generalize, similar to the teacher model. Since (Hinton et al., 2015), numerous KD variants (Romero et al., 2015; Yim et al., 2017; Zagoruyko & Komodakis, 2017; Li et al., 2019; Tian et al., 2020a; Sun et al., 2019) based on feature map, attention map, or contrastive learning have been proposed. (Romero et al., 2015) introduced intermediate-level hints from the teacher model to guide the student model training. (Romero et al., 2015) trained the student model in two stages. First, they trained the student model such that the student's middle layer predicts the output of the teacher's middle layer (hint layer). Next, they fine-tuned the pre-trained student model using the standard KD optimization function. Thanks to the intermediate hint, the student model achieved better performance with fewer parameters. Moving a step further, (Yim et al., 2017), (Zagoruyko & Komodakis, 2017) and (Li et al., 2019) used information from multiple teacher layers to guide students' training. (Yim et al., 2017) computed Gramian matrix between the first and the last layer's output features to represent the flow of problem-solving. (Yim et al., 2017) transferred knowledge by minimizing the distance between student and teacher's flow matrix. (Li et al., 2019) calculated the inter-layered Gramian matrix and inter-class Gramian matrix to find the most representative layer and then minimized the distance between a few of the most representative student and teacher layers. (Zagoruyko & Komodakis, 2017) minimized the distance between teacher and student attention maps at the various block. (Li et al., 2020) distills knowledge from teachers' blocks to supervise students' block-wise architecture search. Contrary to the above methods, which map few teacher-student layers or blocks. We map all student layers to a

teacher layer. We propose RNAS-CL to search for the perfect tutor layer for each student layer. Similar to (Zagoruyko & Komodakis, 2017), we minimize the distance between mapped student-teacher attention maps.

## 2.2 Neural Architecture Search

Neural Architecture Search (NAS) is a technique that automatically designs neural architecture without human intervention. Given a search space, we can find the best architecture by training all architectures from scratch to convergence; however, this is computationally impractical. Earlier studies in NAS were based on RL (Zoph & Le, 2017; Tan et al., 2019) and EA (Real et al., 2017); however, they required lots of computation resources. Most recent studies (Liu et al., 2019; Cai et al., 2019; Wu et al., 2019) encoded architectures as a weight-sharing a super-network. Specifically, they trained an over-parameterized network containing all candidate paths. During training, they introduced weights corresponding to each path. These weights were optimized using gradient descent to select a single network in the end. The selected network is then trained in a standard fashion. Although these methods achieved SOTA performance on various classification tasks, their robustness against adversarial attacks is unknown. (Devaguptapu et al., 2021; Guo et al., 2020; Li et al., 2021; Madry et al., 2018; Su et al., 2018; Xie & Yuille, 2020; Huang et al., 2021) found an intrinsic influence of network architecture on adversarial robustness. (Devaguptapu et al., 2021) observed handcrafted architectures are more robust against adversarial attacks as compared to NAS models. Furthermore, they empirically observed that an increase in model size increased the robustness of the model against adversarial attacks. (Guo et al., 2020) discovered that densely connected architectures are more robust to adversarial attacks. Thus they proposed a NAS method that conducts adversarial training on super-net and then selects the architecture with dense connections. (Li et al., 2021) dilated the backbone network to preserve its standard accuracy and then optimized the architecture and parameters using the adversarial training. Despite SOTA performance, a major drawback lies in the fact that adversarial training is highly time-consuming and decreases the performance on standard (clean) images. This paper proposes a NAS method that optimizes robustness and prediction accuracy without adversarial training.

## 2.3 Efficient and Robust models

Research community has extensively researched building efficient and adversarially robust models individually. However, few works combine both domains, building an efficient and adversarially robust model. (Sehwag et al., 2020) propose to make the pruning technique aware of the robust training objective. They formulate pruning as an empirical risk minimization (ERM) problem and integrate it with a robust training objective. (Huang et al., 2021) investigated the impact of network width and depth configurations on the robustness of adversarial trained DNNs. They observed that reducing capacity at last blocks improves adversarial robustness. (Goldblum et al., 2020), propose Adversarially Robust Distillation (ARD), where they encourage student networks to mimic their teacher's output within an $\epsilon$-ball of training samples. Furthermore, there are few NAS methods (Yue et al., 2022; Ning et al., 2020; Xie et al., 2021) that jointly optimises accuracy, latency and robustness. Compared to these methods, similar-sized RNAS-CL models achieve both higher clean and robust accuracy.

## 3 Robust Knowledge Distillation for Neural Architecture Search

We use knowledge distilled from a robust teacher model to search for a robust and efficient architecture. Knowledge distillation is the transfer of knowledge from a large teacher model to a small student model. In standard knowledge distillation, outputs from the teacher model are used as "soft labels" to train the student model. However, apart from the final teacher outputs, intermediate features constitute important attention information. Different intermediate layers "attend" to different parts of the input object. In RNAS-CL, apart from learning from the teacher's soft labels, the method learns from intermediate teacher layers where to pay attention, i.e., each student layer is mapped to a robust teacher layer to learn where to look. In Section 3.1, we discuss how we define attention maps. We hypothesize that learning where to pay attention from a robust teacher will inherently make the student model more robust to adversarial attacks. Now, the teacher and student could have a different number of layers, which leads us to the question, how to map a student and a teacher layer? In our method, we search for the perfect tutor for each layer. Furthermore, along with increasing

the robustness, we are also interested in searching for an efficient architecture. In Section 3.2 and 3.3, we discuss our tutor and architecture search algorithm. Similar to other state-of-the-art NAS methods (Liu et al., 2019; Wu et al., 2019; Wan et al., 2020), RNAS-CL consists of the searching and training phase. In the search phase, we optimize the architectural weights. In the training phase, we train the architecture sampled from the search phase in a standard fashion. In Section 3.4, we discuss our searching and training optimization objectives.

## 3.1 ATTENTION MAP

We are interested in learning where to pay attention from a robust teacher model. Let us consider a convolution layer with activation tensor $A \in R^{C \times H \times W}$ where $C$ is the number of channels, and $H$ and $W$ are spatial dimensions. We define a mapping function $\mathcal{F} : R^{C \times H \times W} \longrightarrow R^{H \times W}$ that takes $A$ as input and outputs an attention map $\mathcal{F}(A) \in R^{H \times W}$ by $[\mathcal{F}(A)]_{hw} = \sum_{c=1}^{C} A_{c,h,w}^2$, where $A_{c,h,w}$ represents the element of $A$ with channel coordinate $c$ and spatial coordinates $h$ and $w$.

We use activation-based mapping function $\mathcal{F}$ as proposed in (Zagoruyko & Komodakis, 2017). The mapping function $\mathcal{F}$ is applied to activation tensors after each convolution layer to generate an attention map. We visualized few attention maps in Figure 2(b). RNAS-CL intends to find a teacher layer, referred to as a tutor, for each student layer such that the student layer's attention map is similar to that of its tutor in the teacher model. The student attention map may differ in dimension compared to that of its tutor. To address this issue, we interpolate all attention maps to a common dimension.

## 3.2 TUTOR SEARCH

As described above, we aim to find a tutor (teacher layer) for each student layer, which teaches where to pay attention. However, each student layer can choose any tutor, resulting in an exponentially large search space. For example, the search space for a student model with 20 layers and a teacher model with 50 layers is of size $50^{20}$. In order to address the computational issue, we employ Gumbel-Softmax (Jang et al., 2017) to search for the tutor for each student layer in a differentiable manner. Given network parameter $v = [v_1, \ldots, v_n]$ and a constant $\tau$. The Gumbel-Softmax function is defined as $g(v) = [g_1, \ldots, g_n]$ where $g_i = \frac{\exp[(v_i+\epsilon_i)/\tau]}{\sum_i \exp[(v_i+\epsilon_i)/\tau]}$ and $\epsilon_i \sim N(0,1)$ is the uniform random noise, which is also referred to as Gumbel noise. When $\tau \to 0$, Gumbel-Softmax tends to the $\arg\max$ function. Gumbel-Softmax is a "re-parametrization trick", that can be regarded as a differentiable approximation to the argmax function.

Now consider a teacher $T$ and student $S$ model with $n_t$ and $n_s$ number of layers, respectively. $A_t^i$ and $A_s^i$ are the $i^{th}$ activation tensors of teacher and student layers. In RNAS-CL, each student layer $(i)$ is associated with $n_t$ Gumbel weights $(g_i)$ such that $g_i \in R^{1 \times n_t}$. Let $g_{ij}$ be the Gumbel weight associated with $i^{th}$ student and $j^{th}$ teacher layer. Then the attention loss is defined as

$$L_{\text{Attn}}(A_t, A_s) = \frac{1}{n_s \times n_t} \sum_{i=0}^{n_s} \sum_{j=0}^{n_t} g_{ij} \left|\left| \frac{\mathcal{F}(A_s^i)}{||\mathcal{F}(A_s^i)||_2} - \frac{\mathcal{F}(A_t^j)}{||\mathcal{F}(A_t^j)||_2} \right|\right|_2, \tag{1}$$

where $A_s$ and $A_t$ are activation tensors for all student and teacher convolution layers. $\mathcal{F}$ is the mapping function as defined in Section 3.1. $\| \cdot \|_2$ is the $\ell^2$-norm. We exponentially decay the temperature $\tau$ of Gumbel-Softmax while searching, leading to an encoding close to a one-hot vector.

## 3.3 ARCHITECTURE SEARCH

Apart from searching the tutor for each layer, we are interested in building efficient architecture with low latency. Inspired by FBNetV2 (Wan et al., 2020), we search for the optimal number of filters, or the number of output channels, for each convolution block. Let $A = \{f_1, f_2, ..., f_n\}$ be the choices of filters and $\{z_1, z_2, ..., z_n\}$ be their corresponding outputs for a convolution block. Then the cumulative output is defined as $Z = \sum_{i=1}^{n} g_w^{(i)} z_i$, where $g_w^{(i)}$ is the Gumbel weight corresponding to $i^{th}$ filter choice. The number of FLOPs is optimized so as to ensure low latency. The FLOPs are proportional to the number of filters, and the cumulative number of filters is a function of Gumbel weights. As a result, the FLOPs can be optimized in a differential manner using SGD. Similar to

tutor search, temperature is exponentially decayed to obtain an encoding which is close to an one-hot vector. Figure 12 in the appendix illustrates the architecture search process by FBNetV2.

## 3.4 RNAS-CL Loss

Following the convention of state-of-the-art NAS methods (Liu et al., 2019; Wu et al., 2019; Wan et al., 2020), RNAS-CL has searching and training phases. In the search phase, Gumbel weights and other model parameters are updated at each epoch of SGD, where the Gumbel weights correspond to the intermediate student-teacher connection (3.1) and the filter choices (3.3). The weights are optimised using our RNAS-CL search loss defined by (2).

**RNAS-CL search loss.** Let $y$ be the ground-truth one-hot encoded vector, $p$ and $q$ be output probabilities of the student and teacher network and $A_s$, $A_t$ be activation tensors for all student and teacher convolution layers. Then the RNAS-CL search loss is defined as

$$L(y, p, q, A_t, A_s) = (-y \log p + KL(p, q) + \gamma_s L_{\text{Attn}}(A_t, A_s))n_f, \qquad (2)$$

where $KL(p, q) = \sum_i p_i \log \frac{p_i}{q_i}$ is the Kullback–Leibler(KL) divergence between two probability measures. $L_{\text{Attn}}$ is the attention loss as defined in (1) and $\gamma_s$ is a normalization constant. $n_f$ represents latency, which is optimized in a differential manner following (Wan et al., 2020).

After the search phase, a tutor is selected as the $j^*$ teacher layer with $j^* = \arg\max_j g_{ij}$ for each student layer $i$. In addition, the filter choices described in Section 3.3 for neural architecture are decided as the one corresponding to the maximum Gumbel weight for each convolution block. We then start the training phase, where the searched architecture is trained using the RNAS-CL train loss defined below.

**RNAS-CL train loss.** Let $y$ be the ground-truth one-hot encoded vector, $p$ and $q$ be output probabilities of the student and teacher network, and $A_t$, $A_s$ be activation tensors for all student and teacher convolution layers. Then the RNAS-CL train loss is

$$L(y, p, q, A_t, A_s) = L_{\text{CE}}(y, p) + KL(p, q) + \gamma_t L_{Attn}(A_t, A_s), \qquad (3)$$

where $L_{\text{CE}}(y, p) = -y \log p$ is the cross-entropy, $KL(p, q)$ is the KL-divergence, $\gamma_t$ is a normalization constant. Note that, $g_i$ in $L_{Attn}$ is a one-hot vector. Thus, each student attention map is optimized w.r.t. to a single tutor layer.

## 4 EXPERIMENTS

In this section, we conduct experiments on real-world datasets to show the effectiveness of the proposed framework. The experiments section is organized as follows. In Section 4.1, we discuss our experimental setup and implementation details. In Section 4.2, we compare models trained by RNAS-CL against state-of-the-art efficient and robust models on CIFAR-10. In Section 4.3, we empirically show the effectiveness of cross-connections in improving the adversarial robustness of the model. We further discuss the robustness-inducing capacity of teacher layers and compare RNAS-CL models trained on ImageNet-100 in the appendix.

## 4.1 IMPLEMENTATION DETAILS

In this paper, we evaluate RNAS-CL on two public benchmarks for image classification. (1) *CIFAR-10* - a collection of $60k$ images in 10 classes (Krizhevsky, 2009). (2) *ImageNet-100* - a subset of ImageNet-1k dataset (Russakovsky et al., 2015) with 100 classes and about $130k$ images (Tian et al., 2020c). We use standard data augmentation techniques for each dataset, such as random-resize cropping and random flipping. We train different architectures found by RNAS-CL on both CIFAR-10 and ImageNet-100. On each dataset, we first perform the searching step. We train our model using RNAS-CL search loss (2). We search for the channel number and the connected teacher layer at each student layer. We conduct experiments with different search spaces and various robust teacher models. In this section, we refer to our model by *RNAS-CL-X-T* where X represents our search space, and T represents the robust teacher model. Detailed search space is provided in Table 7 and Table 8. We use 4 robust teacher model, ResNet-50, ResNet-18, WideResNet-50, and WideResNet-34, which are referred to as R-50, R-18, WRT-50, and WRT-34. For example, RNAS-CL-S3-R-18 represents a model trained in the S3 search space using an adversarially robust ResNet-18 model.

| Method | Clean Acc | FSGM | PGD$^{20}$ | MI-FGSM | # Params (M) | MACs (M) |
|---|---|---|---|---|---|---|
| Without Adversarial Training | | | | | | |
| DARTS (Liu et al., 2019) | 97.03 | 42.48 | 7.09 | 0.28 | 3.3 | 500* |
| PC-DARTS (Xu et al., 2020) | 97.05 | 49.18 | 9.84 | 1.21 | 3.6 | 600* |
| RACL (Dong et al., 2020) | 97.44 | 50.53 | 1.93 | 4.68 | 3.6 | 500* |
| AmoebaNet (Real et al., 2019) | 97.39 | 44.79 | 0.25 | 0.80 | 3.2 | 500* |
| NasNet (Zoph et al., 2018) | 97.37 | 47.53 | 0.42 | 1.01 | 3.8 | 600* |
| MVV2-ARD (Goldblum et al., 2020) | 76.13 | - | 38.21 | - | 3.4 | 300 |
| E2RNAS-C16 (Yue et al., 2022) | 93.97 | - | 6.76 | - | 0.44 | - |
| RNAS-CL-S3-WRT-34 (Our) | **89.4** | 44.95 | **34.3** | 38.92 | **0.11** | **6.64** |
| RNAS-CL-S5-WRT-34 (Our) | 90.4 | 46.72 | 35.59 | 40.57 | 0.21 | 11.02 |
| RNAS-CL-S7-WRT-34 (Our) | 90.62 | 48.93 | 37.24 | 42.27 | 0.32 | 15.58 |
| RNAS-CL-M-WRT-34 (Our) | 92.46 | 50.51 | 39.84 | 44.54 | 3 | 326 |
| RNAS-CL-L–WRT-34 (Our) | **92.6** | 52.37 | **41.9** | **46.66** | 11 | 1210 |
| With Adversarial Training | | | | | | |
| Hydra ResNet 18 (Sehwag et al., 2020) | 69 | - | 41.6 | - | 0.11 | 37.63 |
| Hydra ResNet 34 (Sehwag et al., 2020) | 71.8 | - | 44.4 | - | 0.21 | 75.43 |
| Hydra ResNet 50 (Sehwag et al., 2020) | 73.9 | - | 45.3 | - | 0.25 | 85.92 |
| ADV-ADMM ResNet 18 (Ye et al., 2019) | 58.7 | - | 36.1 | - | 0.11 | 37.63 |
| ADV-ADMM ResNet 34 (Ye et al., 2019) | 68.8 | - | 41.5 | - | 0.21 | 75.43 |
| ADV-ADMM ResNet 50 (Ye et al., 2019) | 69.1 | - | 42.2 | - | 0.25 | 85.92 |
| RobNet-Small (Guo et al., 2020) | 78.05 | 53.93 | 48.32 | 48.98 | 4.41 | - |
| RobNet-Medium (Guo et al., 2020) | 78.33 | 54.55 | 49.13 | 49.34 | 5.66 | - |
| RobNet-Large (Guo et al., 2020) | 78.57 | 54.98 | 49.44 | 49.92 | 6.89 | - |
| AmoebaNet (Real et al., 2019) | 83.41 | 56.40 | 39.47 | 47.60 | 3.2 | 500* |
| NasNet (Zoph et al., 2018) | 83.66 | 55.67 | 48.02 | 53.05 | 3.8 | 600* |
| DARTS (Liu et al., 2019) | 83.75 | 55.75 | 44.91 | 51.63 | 3.3 | 500* |
| PC-DARTS (Xu et al., 2020) | 83.94 | 52.67 | 41.92 | 49.09 | 3.6 | 600* |
| RACL (Dong et al., 2020) | 83.89 | 57.44 | 49.34 | 54.73 | 3.6 | 500* |
| VGG-11-R (Huang et al., 2021) | 79.63 | 57.35 | 43.93 | - | 5.83 | - |
| DN-121-R (Huang et al., 2021) | 87.22 | **67.12** | 52.52 | - | 6 | - |
| DARTS-R (Huang et al., 2021) | 87.2 | 66.74 | 52.36 | - | 2.53 | - |
| MVV2-ARD (Goldblum et al., 2020) | 84.70 | - | 46.28 | - | 3.4 | 300 |
| MSRobNet-1000 (Ning et al., 2020) | 84.5 | 59.6 | 52.7 | - | 3.16 | - |
| MSRobNet-2000 (Ning et al., 2020) | 85.7 | 60.6 | 53.6 | - | 6.46 | - |
| S$^2_{8/255}$ (Xie et al., 2021) | 76.54 | - | 31.83 | - | 1.68 | - |
| RNAS-CL-S3-WRT-34 (Our) | 83.45 | 50.67 | 43.07 | 43.98 | **0.11** | **6.64** |
| RNAS-CL-S5-WRT-34 (Our) | 84.75 | 51.99 | 44.68 | 46.3 | 0.21 | 11.02 |
| RNAS-CL-S7-WRT-34 (Our) | 85.81 | 49.11 | 43.24 | 45.53 | 0.32 | 15.58 |
| RNAS-CL-M-WRT-34 (Our) | **87.29** | 59.71 | **51.76** | 53.43 | 3 | 326 |
| RNAS-CL-L-WRT-34 (Our) | 86.28 | 61.12 | **53.69** | **55.07** | 11 | 1210 |

Table 1: The table shows performance of various efficient and robust methods on CIFAR-10 dataset. Clean Acc represents top-1 accuracy on clean images. FSGM, PGD$^{20}$, MI-FGSM represents top-1 accuracy on images perturbed by the corresponding attacks. PGD$^{20}$ represents 20 step PGD attack. $*$ represents approximate values. Columns with unreported values are represented by -.

For both datasets, we use SGD optimizer. For ImageNet-100, default values of momentum and weight decay are set to $0.9$ and $4e − 5$, respectively. The batch size is set to $256$. The learning rate is initialized as $0.05$ and annealed down to zero following a cosine schedule. After the search stage which takes 100 epochs, the searched architecture is trained from scratch using RNAS-CL train loss (3) for 200 epochs. For CIFAR-10, default values of momentum and weight decay are set to $0.9$ and $2e − 4$, respectively. The batch size is set to $128$. We train our model for 100 epochs in both the searching and training phases. The learning rate is initialized as $0.1$, and reduced by a factor of $10$ after the $75^{th}$ and the $90^{th}$ epoch. Following the settings of FBNetV2, the temperature ($\tau$) in Gumbel-Softmax is initialized as $5.0$ and exponentially annealed by $e^{−0.045}$ every epoch in the search phase. The hyper-parameter $\lambda_s$ and $\lambda_t$ in (2, 3) is selected from a candidate set $\{0.01, 0.1, 0.1, 1.0, 10, 100\}$. Both $\lambda_s$ and $\lambda_t$ are set to $1.0$ for all experiments. In the search phase for each batch, we use 80% of the data to optimize the model weights and the remaining 20% data to optimize architectural weights which are Gumbel weights. For robustness evaluation, we choose five powerful attacks including FGSM (Goodfellow et al., 2015), MI-FGSM (Dong et al., 2018), PGD (Madry et al., 2018), CW (Carlini & Wagner, 2017) and AutoAttack (Croce & Hein, 2020). Results for CW and AutoAttack are provided in the appendix A.4. Consistent with adversarial literature (Madry et al., 2018; Zhang et al., 2019b), the adversarial perturbation is considered under $l_\infty$ norm with a total perturbation scale of 8/255 (0.031).

## 4.2 COMPARE EFFICIENT AND ROBUST CIFAR-10 MODELS

In this section, we compare the robustness of our method against other SOTA efficient and robust models. In Table 1, we compare RNAS-CL to both efficient models trained with and without adversarial training. All RNAS-CL models are trained with robust WideResNet-34 (Rice et al., 2020) as the teacher model. RNAS-CL significantly outperforms all models trained without adversarial

training in terms of adversarial accuracy. While being significantly smaller, our models achieve significantly higher adversarial accuracy when compared to models trained without adversarial training. For example, *RNAS-CL-S7-WRT-34* achieves more than $28\%$ higher PGD accuracy compared to most of the other methods. Compared to MVVV2-ARD, *RNAS-CL-S7-WRT-34* achieves $\sim 1\%$ lower PGD accuracy; however, it exceeds MVVV2-ARD by $14.5\%$ in clean accuracy while being $10\times$ smaller. A similar-sized model, for example, *RNAS-CL-M-WRT-34*, exceeds both clean and PGD accuracy by $16.5\%$ and $1.43\%$.

Next, we compare RNAS-CL against adversarially trained robust models. For a fair comparison, after the training stage, we train our RNAS-CL models with the TRADES optimization objective for 20 epochs. For retraining, the cross-entropy term in (3) is replaced by TRADES optimization objective. Adversarially training RNAS-CL models improve its adversarial accuracy. RNAS-CL models achieve similar or higher adversarial accuracy compared to other adversarially trained models. However, RNAS-CL models are much smaller and achieve significantly higher clean accuracy. For example, in Table 1, RNAS-CL-M-WRT-34 achieves similar or higher adversarial accuracy than most other methods while being smaller and significantly exceeding in terms of clean accuracy. We also obtain much smaller models using RNAS-CL. Tiny RNAS-CL models exceed their counterpart by more than $\sim 12\%$ in terms of clean accuracy. For example, RNAS-CL-S5-WRT-34 exceeds HYDRA (ResNet-34) by $12.95\%$ while achieving similar adversarial accuracy. Similar results can also be visualized in Figure 1. In Figure 1, RNAS-CL models are on the top right corner of the plot, representing the models with the highest clean and adversarial accuracy. Results for RNAS-CL models trained with different robust teachers have been added to the appendix A.2.

**Comparison against various perturbation budget** To further illustrate the effectiveness of RNAS-CL, we compare RNAS-CL with previously proposed defense mechanisms against various perturbation budgets. In Figure 3, we compare various methods against PGD and FSGM attacks. For both attacks, RNAS-CL outperforms its counterparts at all perturbations. RNAS-CL significantly outperforms other methods as perturbation size increases. For $\epsilon = 0.1$, RNAS-CL exceeds other methods by $\sim 20\%$ for both PGD and FSGM attacks.

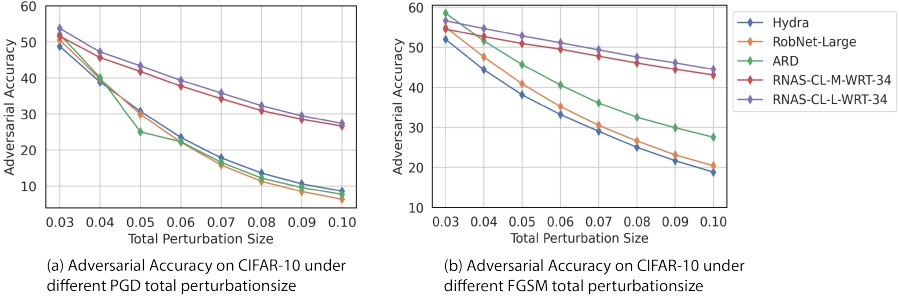

(a) Adversarial Accuracy on CIFAR-10 under different PGD total perturbationsize

(b) Adversarial Accuracy on CIFAR-10 under different FGSM total perturbationsize

Figure 3: Robustness evaluation under different perturbation sizes for PGD and FGSM attacks.

### 4.3 ABLATION STUDY

This ablation study demonstrates the significance of student-teacher cross-layer connections in RNAS-CL. We compare four types of training paradigms. In the first training paradigm, we conduct searching and training using cross-entropy loss without any teacher model. We refer to this as standard. Next, in the second paradigm, we conduct searching and training by minimizing the cross-entropy loss and standard KL Divergence with a robust teacher model. We refer to them as KL-X-T, where X represents the search space and T represents the robust teacher model. In the third paradigm we search and train using cross-entropy loss and intermediate cross connections (ICC). We refer to them as ICC-X-T. Finally, the fourth model type is RNAS-CL, where we include all three terms, cross-entropy loss, KL Divergence, and cross-layer student-teacher connections.

In Figure 4(a), we compare the attention maps from student models trained using RNAS-CL-I-R-50 against students trained using KL-I-R-50. We compare attention maps for various convolution layers at regular intervals. As expected, adding cross-layer connections obtains attention maps from the student model closer to the teacher model. Each student layer learns where to pay attention from its connected teacher layer. For example, in column (b), the KL-I-R-50 layer attends to various parts of the image, whereas the RNAS-CL layer learning from $28^{th}$ teacher layer pays more attention

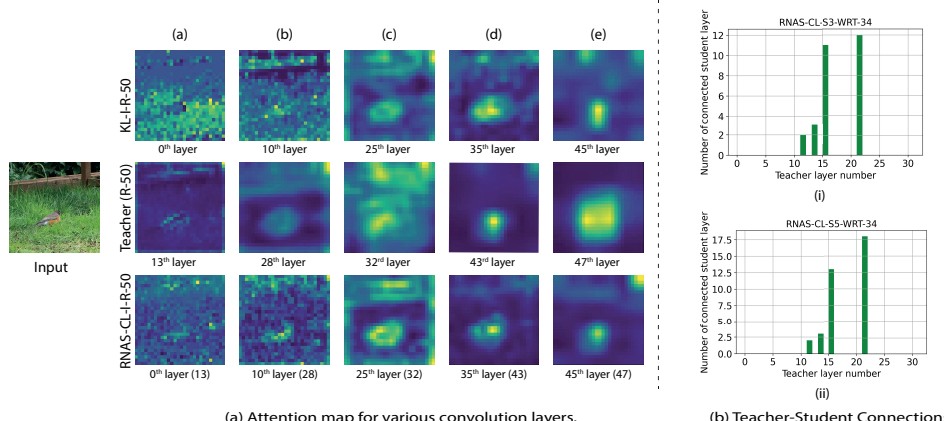

(a) Attention map for various convolution layers.       (b) Teacher-Student Connections

Figure 4: (a) KL-I-R-50 represents attention maps from a model trained using cross-entropy loss and knowledge distillation without any cross-layer connections. Teacher and RNAS-CL represent attention maps from the robust teacher (ResNet-50) and RNAS-CL model. Name for each RNAS-CL layer includes its connected teacher layer. For example, in $0^{th}$ layer (13), 13 represents the corresponding teacher layer. RNAS-CL drives attention maps from student layers closer to their corresponding teacher layer.(b) Illustrations of the number of student layers connected to each teacher layer in RNAS-CL for various student models on the CIFAR-10 dataset.

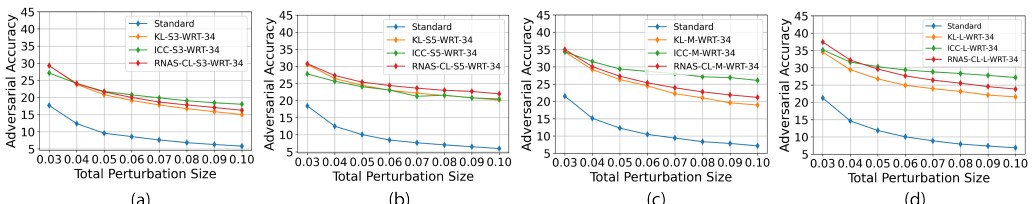

Figure 5: Adversarial accuracy of various models at various perturbation budgets.

central part of the image. Similarly, in column (c), the RNAS-CL layer learns from the teacher model to pay more attention to the central and upper portions of the image. In Figure 5, we compare RNAS-CL models against KL-X-T and standard models against PGD attacks at various perturbation budgets on the CIFAR-10 dataset. The RNAS-CL and ICC models outperform their counterparts, demonstrating the significance of cross-connections.

**Robust Teacher layers** We hypothesize that few teacher layers are more robust than others and thus should induce more robustness to the student models. In RNAS-CL, each student layer is associated with a teacher layer. Figures 4(b) illustrates the number of student layers connected to each robust teacher layer on the CIFAR-10 dataset. For both student models, we observe that $15^{th}$ and $21^{st}$ layers of the robust teacher model have significantly more intermediate connections, suggesting that few teacher layers have more robustness-inducing capacity than others. Plots for other student models have been added to the appendix A.1.

## 5    CONCLUSION

In this paper, we propose Robust Neural Architecture Search by Cross-Layer Knowledge Distillation (RNAS-CL), a novel NAS algorithm that improves the robustness of the student model by learning from a robust teacher through cross-layer knowledge distillation. RNAS-CL optimizes neural architecture to achieve a good tradeoff between robustness and clean accuracy in a differentiable manner without robust training. RNAS-CL extends conventional knowledge distillation by learning student-teacher cross-connections. We show that models obtained by RNAS-CL outperform all models obtained without robust training in terms of adversarial robustness. We show that adding adversarial training can further increase the adversarial robustness of RNAS-CL models. After robust training, RNAS-CL achieves similar adversarial robustness compared to models obtained via robust training while outperforming them in terms of clean accuracy. For future work, we plan to incorporate robust training in the searching stage to further increase the robustness of the model.

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

# A APPENDIX

## A.1 ROBUST TEACHER LAYERS

In this section, we discuss robustness inducing capacity of teacher layers. We hypothesize that few teacher layers are more robust than others and thus should induce more robustness to the student models. In RNAS-CL, each student layer is associated with a teacher layer. Figures 6 and 8 plot the number of student layers connected to each robust teacher layer on the CIFAR-10 and ImageNet-100 datasets. For all student models on CIFAR-10, we observe that layers 15 and 21 of the robust teacher model have significantly more intermediate connections with the student models. Similarly, for ImageNet-100, layers 18, 32, and 40 are a few of the dominant robust layers. In Figures 7 and 9, we visualize the most robust teacher layers on CIFAR-10 and ImageNet, respectively.

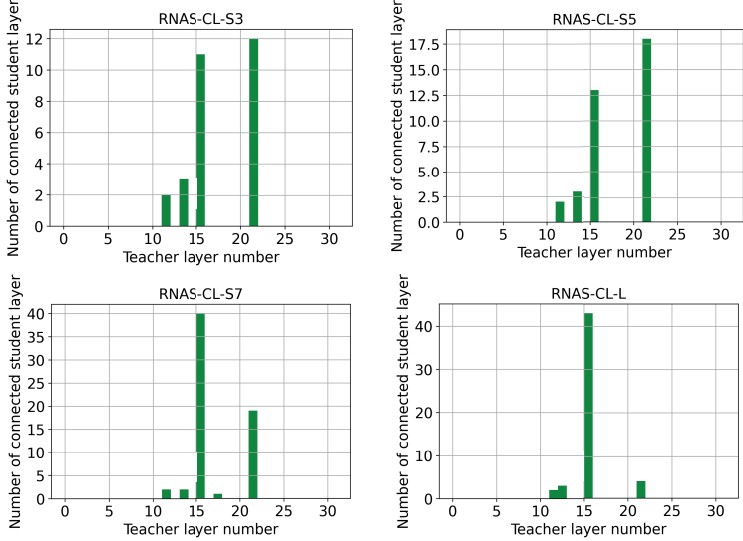

Figure 6: Illustrations of the number of student layers connected to each teacher layer in RNAS-CL for various student models on the CIFAR-10 dataset. We choose adversarially trained Wide-ResNet-34 as the robust teacher model for all the four student models, with one plot for each student model. All student architectures are described in Table 7.

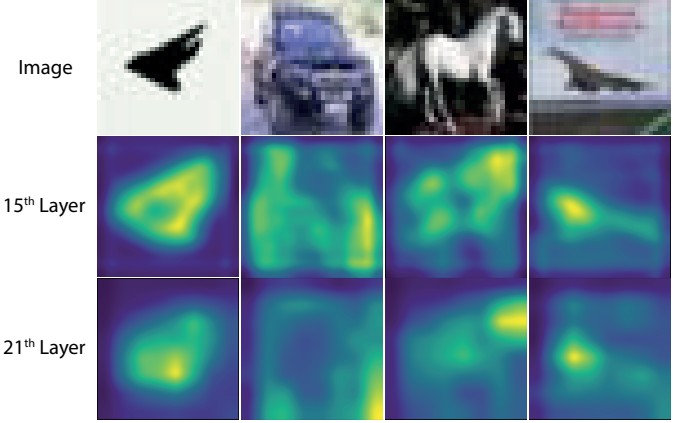

Figure 7: Attention map for most robust teacher layers on CIFAR-10 dataset. We chose the same robust teacher model as in Figure 6. The illustrated layers represent teacher layers with maximum number of intermediate connection for various RNAS-CL models (as described in Figure 6).

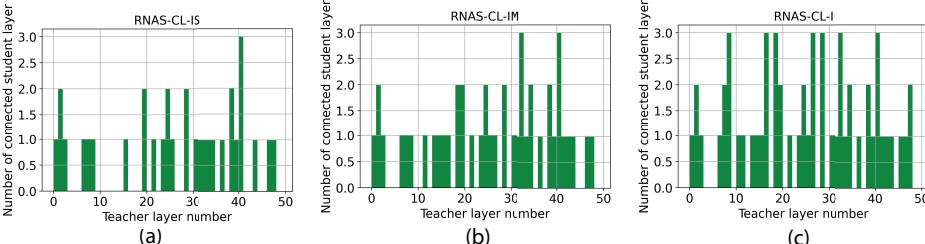

Figure 8: Illustrations of the number of student layers connected to each teacher layer in RNAS-CL for various student models on the ImageNet-100 dataset. We choose adversarially trained Wide-ResNet-50 as the robust teacher for all and three students models, with one plot for each student model. All RNAS-CL architectures are described in Table 8.

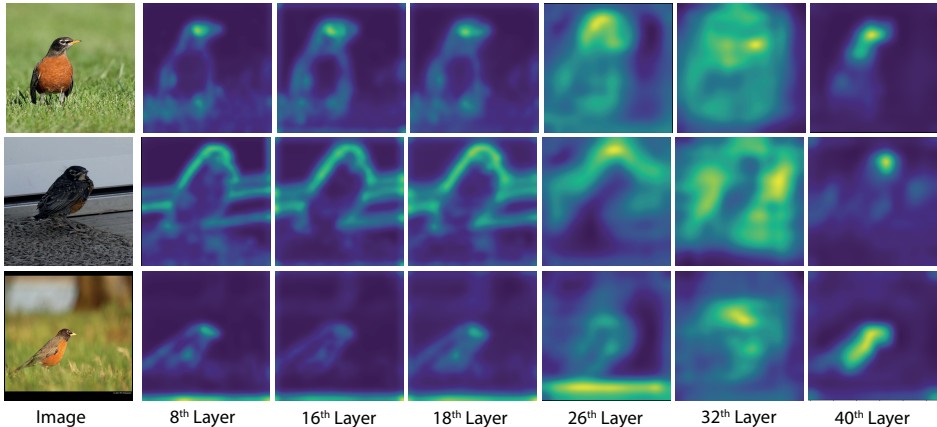

Figure 9: Attention maps for most robust teacher layers on ImageNet-100 dataset. We chose the same robust teacher model as in Figure 8. The illustrated layers represent teacher layers with maximum number of intermediate connection for various RNAS-CL models (as described in Figure 8).

## A.2 MORE RESULTS ON CIFAR-100

In this section, we conduct experiments using adversarially trained WRT-34 (Rice et al., 2020), ResNet-50 (Engstrom et al., 2019), and ResNet-18 (Sehwag et al., 2021) as the robust teacher models on the CIFAR-10 dataset. All RNAS-Cl models, while achieving similar clean accuracy, exceed its counterpart by more than 10% in PGD accuracy. RNAS-CL-R50 achieves higher robust accuracy than RNAS-CL-R18 and RNAS-CL-WRT-34. However, ResNet-50 has the lowest PGD accuracy among the teacher models, suggesting that the teacher's architecture has more influence on the student's performance than the teacher's performance. The higher number of teacher layers allows more options for the student layer to learn from, leading to better robustness. The teacher models' performance is reported in Table 6.

## A.3 COMPARE EFFICIENT AND ROBUST IMAGENET-100 MODELS

We compare RNAS-CL to adversarially robust pruning methods on ImageNet-100 dataset, with results shown in Table 3. RNAS-CL models are trained with three different robust teachers, ResNet-18, ResNet-50, and WideResNet-50, with the ImageNet pre-trained (Engstrom et al., 2019) being the robust teacher. It is observed that RNAS-CL models consistently exceed other models by $\sim$ 25% in terms of clean accuracy while exibiting adversarial robustness. In Table 3, both Hydra and LWM were adversarially trained using TRADES (Zhang et al., 2019a). For a fair comparison, after the regular training stage without TRADES, we retrain our RNAS-CL models with the TRADES optimization objective. We replace the cross-entropy term in (3) by the TRADES optimization

| Method | Clean | PGD$^{20}$ |
|---|---|---|
| Standard-S3 | 89.92 | 17.69 |
| Standard-S5 | 90.76 | 18.44 |
| Standard-S7 | 90.98 | 19.3 |
| RNAS-CL-S3-WRT-34 | 89.4 | 34.3 |
| RNAS-CL-S5-WRT-34 | 90.4 | 35.59 |
| RNAS-CL-S7-WRT-34 | 90.62 | 37.24 |
| RNAS-CL-S3-R50 | 89.39 | 35.76 |
| RNAS-CL-S5-R50 | 90.53 | 37.32 |
| RNAS-CL-S7-R50 | 90.41 | 37.98 |
| RNAS-CL-S3-R18 | 88.47 | 26.35 |
| RNAS-CL-S5-R18 | 88.77 | 25.49 |
| RNAS-CL-S7-R18 | 89.47 | 27.96 |

Table 2: Performance of RNAS-CL method trained with various robust teacher models on the CIFAR-10 dataset. Standard represents models searched and trained by cross-entropy loss without any teacher model.

| Method | Clean | PGD$^{20}$ | # Params (M) | MACs (M) |
|---|---|---|---|---|
| Hydra (ResNet-18) - 90% (Sehwag et al., 2020) | 59.96 | 29.79 | **1.1** | 1200 |
| LWM (ResNet-18) - 90% (Han et al., 2015) | 59.02 | 27.67 | 1.1 | 1200 |
| RNAS-CL-I-R-18 | 85.22 | 8.3 | 3.94 | **241.98** |
| RNAS-CL-I-R-50 | **85.98** | 5.08 | 3.96 | 244.76 |
| RNAS-CL-I-WRT-50 | 85.46 | 3.36 | 4.01 | 255.37 |
| RNAS-CL-I-R-18 + TRADES | 78.94 | 29.02 | 3.94 | **241.98** |
| RNAS-CL-I-R-50 + TRADES | 79.95 | **32.44** | 3.96 | 244.76 |
| RNAS-CL-I-WRT-50 + TRADES | 79.42 | 28.06 | 4.01 | 255.37 |

Table 3: Performance of various efficient and robust methods on ImageNet-100 dataset. Clean Acc and Adv Acc are the same as that in Table 1. All MACs were calculated without special hardware (Han et al., 2016) or special software (Park et al., 2017)

objective. With such training, RNAS-CL achieves similar or higher adversarial accuracy while significantly outperforming Hydra and LWM in clean accuracy with only a fraction of MACs.

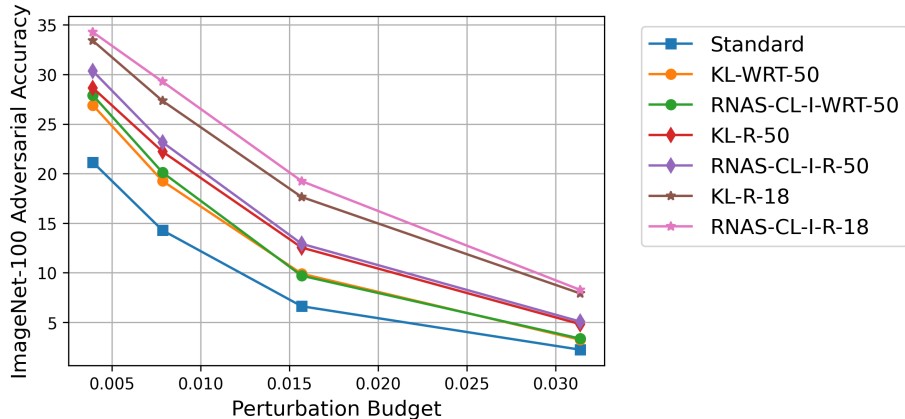

Figure 10: Adversarial accuracy of various models at various perturbation budgets on the ImageNet-100 dataset.

We further study adversarial accuracy at various perturbation budgets for three different teacher models. As illustrated in Figure 10, RNAS-CL exceeds its counterpart in adversarial accuracy at various perturbation budgets for all teacher models on the ImageNet-100 dataset. This demonstrates the significance of cross-layer connections in RNAS-CL.

### A.4 Compare CIFAR-10 model against CW and AutoAttack

In this section, we compare RNAS-CL and (Huang et al., 2021) against recent attacks such as $CW_\infty$ (Carlini & Wagner, 2017) and AutoAttack (Croce & Hein, 2020) on CIFAR-10 dataset. CW attacks were proposed to defeat defensive distillation. In Table 4, we use $L_\infty$ version of CW attack optimized by PGD, with maximum perturbation budget set to $\epsilon = 8/255$. AutoAttack is a parameter-free ensemble attack currently considered one of the most reliable and widely acknowledged evaluation benchmark in Adversarial Defences.

| Method | $CW_\infty$ | AA |
|---|---|---|
| VGG-R (Huang et al., 2021) | 46.49 | 38.44 |
| DN-121-R (Huang et al., 2021) | 53.07 | 47.75 |
| RNAS-CL-S3-WRT-34(Our) | 47.07 | 37.17 |
| RNAS-CL-S5-WRT-34(Our) | 48.33 | 39.28 |
| RNAS-CL-S7-WRT-34(Our) | 47.91 | 38.36 |
| RNAS-CL-M-WRT-34(Our) | **53.52** | 46.89 |
| RNAS-CL-L-WRT-34(Our) | 52.63 | **48.49** |

Table 4: The table compared performance of (Huang et al., 2021) and RNAS-CL against $CW_\infty$ (Carlini & Wagner, 2017) and AutoAttack (Croce & Hein, 2020) on CIFAR-10 dataset.

### A.5 Comparison against KD Varients

In this section, we compare our methods against various knowledge distillation methods Park et al. (2019); Ahn et al. (2019); Tung & Mori (2019); Tian et al. (2020b); Passalis & Tefas (2018). We use Robust WRT-34 as the teacher model for all KD methods and train three different student architectures: RNAS-CL-S3, RNAS-CL-S5, and RNAS-CL-S7. In Figure 11, models trained using our paradigm are explicitly on the upper right-most part of the graph. RNAS-CL-S3 architecture trained using RKD performs similarly to the model trained using our methods. Apart from this, all models trained using RNAS-Cl significantly outperform all other methods in terms of clean and adversarial accuracy.

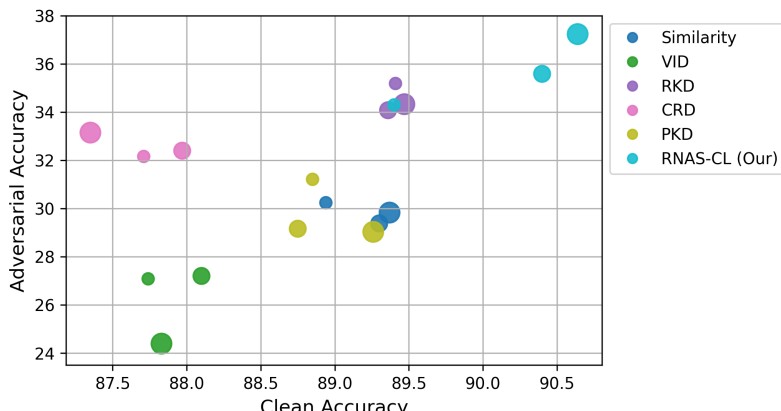

Figure 11: The figure compares various knowledge distillation variants (Similarity (Tung & Mori, 2019), VID (Ahn et al., 2019), RKD (Park et al., 2019), CRD (Tian et al., 2020b), PKD (Passalis & Tefas, 2018)) against RNAS-CL on the CIFAR-10 dataset. Adversarial Accuracy represents top-1 Accuracy on images perturbed by 20 step PGD attack. Clean Accuracy represents top-1 Accuracy on clean images. Larger marker size indicates larger architecture. For each method, RNAS-CL-S3, RNAS-CL-S5, and RNAS-CL-S7 are represented by increasing marker size.

### A.6 Results for ImageNet

In this section, we compare our method on the ImageNet dataset. Standard represents the model searched and trained using cross-entropy loss without any teacher model. RNAS-Cl represents the

model trained using our training paradigm. Both models are further adversarially trained using FastAT (Wong et al., 2020). In Table 5, we evaluate the robustness against 10 step PGD attack with $\epsilon = 4/255$. Models trained with RNAS-CL exceed both in terms of clean and robust accuracy.

| Method | Clean | PGD[10] |
|---|---|---|
| Standard | 53.92 | 25.45 |
| RNAS-CL-WRT-50 | 56.1 | 29.78 |

Table 5: Robustness results on ImageNet dataset.

## A.7 ROBUST TEACHER MODELS

In this section, we report the robustness of adversarially trained teacher model used throughout the paper on CIFAR-10 dataset.

| Model | Clean | PGD[20] |
|---|---|---|
| WRT-34 | 86.07 | 58.33 |
| ResNet 18 | 84.59 | 55.54 |
| ResNet 50 | 87.03 | 49.25 |

Table 6: Robustness results for various teacher model on CIFAR-10 dataset.

## A.8 ARCHITECTURE

In this section, we discuss architectures for various proposed super-nets used in RNAS-CL for CIFAR-10 and ImageNet-100 datasets. Table 7 describes the super-nets used for CIFAR-10. We use super-nets with three blocks. Super-nets used for ImageNet-100 are described in Table 8. For ImageNet-100, the number of blocks varies from 3 to 5.

| Search Space for CIFAR-10 | | | | |
|---|---|---|---|---|
| Search Space | Depth | Stage 1 | Stage 2 | Stage 3 |
| RNAS-CL-S3 | 3-3-3 | 16, 12 | 32, 28, 24, 20 | 64, 60, 56, 52 |
| RNAS-CL-S5 | 5-5-5 | 16, 12 | 32, 28, 24, 20 | 64, 60, 56, 52 |
| RNAS-CL-S7 | 7-7-7 | 16, 12 | 32, 28, 24, 20 | 64, 60, 56, 52 |
| RNAS-CL-M | 9-7-1 | 80, 76 | 160, 156, 152, 148 | 128, 124, 120, 116 |
| RNAS-CL-L | 9-7-1 | 160, 156 | 320, 316, 312, 308 | 256, 252, 248, 244 |

Table 7: The table describes the search space for CIFAR-10. Depth represents the depth of each stage. For example, 3-3-3 represents three convolution blocks in each stage. All search spaces have three stages. Stage 1, Stage 2, and Stage 3 represent the filter choices for their respective stages. For example, at stage 3 of RNAS-CL-S3, for each convolution block, we search between 4 output channels (64, 60, 56, 52).

## A.9 ARCHITECTURE SEARCH BY FBNETV2

RNAS-CL builds both an efficient and adversarially robust deep learning model. In this work, we use the training paradigm of FBNetV2 to search for efficient models. In Figure 12, we illustrate the searching process for neural architecture at a single convolution layer. Each filter choice is attached with a Gumbel weight. These Gumbel weights are optimized to select an efficient model.

| Search Space for ImageNet-100 | | | | | | |
|---|---|---|---|---|---|---|
| Search Space | Depth | Stage 1 | Stage 2 | Stage 3 | Stage 4 | Stage 5 |
| RNAS-CL-IS | 3-3-3 | 28, 24, 20, 16 | 40, 36, 32, 28 | 96, 88, 80, 72, 64, 56, 48 | | |
| RNAS-CL-IM | 3-3-3-4 | 28, 24, 20, 16 | 40, 36, 32, 28 | 96, 88, 80, 72, 64, 56, 48 | 128 120, 108, 100, 92, 84, 76, 68 | |
| RNAS-CL-I | 3-3-3-4-4 | 28, 24, 20, 16 | 40, 36, 32, 28 | 96, 88, 80, 72, 64, 56, 48 | 128 120, 108, 100, 92, 84, 76, 68 | 216, 208, 200, 192, 184,176, 168, 160, 152, 144,136, 128, 120, 108 |

Table 8: The table describes the search space for ImageNet-100. Similar to Table 7, depth represents the depth of each stage. For ImageNet-100, we have up to 5 stages. Stage 1, Stage 2, Stage 3, Stage 4, and Stage 5 represent the filter choices for their respective stages. For example, in stage 1, for each convolution block, we search for its channel within 4 output channel options $(28, 24, 20, 16)$.

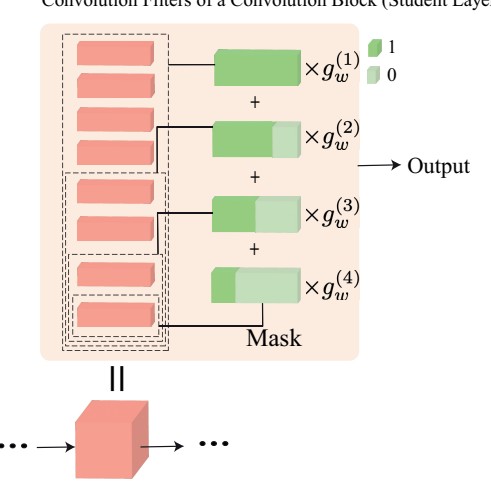

Figure 12: Illustration of searching for the neural architecture of each layer of student model using the searching mechanism in FBNetV2. $g_w^i$ represents gumbel weights associated with each mask.

