# OpenReview forum: "RNAS-CL: Robust Neural Architecture Search by Cross-Layer Knowledge Distillation"
_ICLR.cc/2023/Conference — Submitted to ICLR 2023_

### Official Review · Reviewer_4K2f · 2022-10-24

**Confidence:** 4
**Correctness:** 3
**Technical Novelty And Significance:** 2
**Empirical Novelty And Significance:** 2
**Recommendation:** 5

**Clarity, Quality, Novelty And Reproducibility:**

The paper is well written and easy to understand. The novelty of the proposed method seems to be limited.

**Strength And Weaknesses:**

Strength:
1. It make senses to automatically search for the best teacher layer to supervise each student layer, and ablation studies justify the effectiveness of this strategy.
2. The paper is well written and easy to understand.

Weaknesses:
1. The knowledge distillation method is quite standard except for the optimal teacher layer searching. For example, there are not much new things in Section 3.1 and distillation loss in Section 3.4.
2. The architecture search method seems to have limited novelty over FBNetv2.
3. The proposed method use knowledge distillation to improve the performance, while the compared methods not. It is not quite fair and no ablation studies to show the performance of the proposed method without knowledge distillation.

**Summary Of The Paper:**

This paper proposes a method to obtain a robust and efficient neural architecture by searching the best teacher layers and the number of filters for student network. Experiments are conducted on CIFAR and ImageNet-100, which show good performance over prior methods.

**Summary Of The Review:**

My main concerns lie on the limited novelty of the proposed method and the unfair comparisons with prior work.

---

> ### Author Response · Authors · 2022-11-12
> **Response to Reviewer 4K2f**
>
> We would like to thank the reviewer for his comments on the paper. Below, we address all the points raised by the reviewer. We hope that the reviewer will reconsider the evaluation in light of our responses.
>
> RNAS-CL outperforms the current state-of-the-art efficient and robust models, and it optimizes robustness, prediction accuracy, and latency without robust training. Our experimental results demonstrate that RNAS-CL outperforms the current state-of-the-art efficient and robust models. Without robust training, RNAS-CL exceeds its counterparts by more than 20-30% in terms of robust accuracy. With robust training, RNAS-CL achieves even higher clean and robust accuracy.
>
> Apart from the empirical results, we demonstrate that a few teacher layers are more robust than others. In Figure 4 and 6, we observe that 3-4 teacher layers have significantly more intermediate connections, suggesting that a few teacher layers have more robustness-inducing capacity than others.
>
>
> In Table 1, MVV2-ARD uses knowledge distillation to improve the performance of the student models. The students mimic their teacher's output within an ε -ball of training samples. Furthermore, we compare our method against various knowledge distillation variants in Figure 11. We use Robust WRT-34 as the teacher model for all KD methods and train three student architectures: RNAS-CL-S3, RNAS-CL-S5, and RNAS-CL-S7. All RNAS-CL models have higher adversarial and clean accuracy compared to other KD variants, evidencing the effectiveness of cross-connections.
>
> We have updated Figure 5 with the model's performance without standard knowledge distillation. We refer to this model by Intermediate Cross-Connection (ICC). For all architectures, ICC performs similarly to RNAS-CL and KL models at low perturbations. However, as perturbation size increases, ICC performs significantly better than both models, demonstrating the significance of teacher-student intermediate cross-connections.

---

### Official Review · Reviewer_k7LD · 2022-10-25

**Confidence:** 3
**Clarity, Quality, Novelty And Reproducibility:** See Strength And Weaknesses
**Correctness:** 3
**Technical Novelty And Significance:** 2
**Empirical Novelty And Significance:** 2
**Recommendation:** 5

**Strength And Weaknesses:**

Strength:

-It is interesing to find robust architecture with automatic search.

-The proposed method use a teacher to enhance the model's performance and robustness, which is reasonable.

-This paper is well-written and easy to follow.

Weakness:

-Directly combining NAS and knowledge distillation can usually improve performance. The authors should compare the proposed method with other knowledge distillation methods.

- Only tiny datasets are used to evaluate the effectiveness of the proposed method. These datasets have a large gap with the practical scense. Comparesion on large dataset (such as ImageNet) is required to validate the method's effectiveness.

**Summary Of The Paper:**

This paper proposes a NAS method to find robust neural network that can defend adversarial attracks. It combine knowledge distillation with NAS to pursue better performacne. The authors conduct experiments on multiple tiny datasets, such as CIFAR-10, ImageNet-100.

**Summary Of The Review:**

See Strength And Weaknesses

---

> ### Author Response · Authors · 2022-11-12
> **Response to Reviewer k7LD**
>
> We would like to thank the reviewer for his comments on the paper. We address all the concerns raised by the reviewer below. We hope that the reviewer will reconsider the evaluation in light of our responses.
>
> We compared our methods against various knowledge distillation variants in Figure 11 in the revised paper. We used Robust WRT-34 as the teacher model for all KD methods and trained three student architectures: RNAS-CL-S3, RNAS-CL-S5, and RNAS-CL-S7. All RNAS-CL models have higher adversarial and clean accuracy compared to other KD variants illustrating the effectiveness of cross-connections.
>
> In Table 5 of the revised paper, we conducted our experiments on the ImageNet dataset. The model trained using RNAS-CL exceeded the standard model by 4.3% in terms of robust accuracy.

---

> > ### Author Response · Authors · 2022-11-23
> > **More experiments on ImageNet**
> >
> > We conducted more experiments on the ImageNet dataset. We compare our model against the SOTA compact and efficient method [1], which is known to have the best PGD accuracy (by a compact and efficient method) on ImageNet so far. RNAS-CL significantly outperforms [1] in all three attributes: clean accuracy, robust accuracy, and the number of parameters, as shown in the table below. This result will be added to the final version of this paper.
> >
> > |Method                                |Clean Acc.  |PGD$^{10}$   |#Params|
> > | -------------------------------| ------------| ------------|---------|
> > |ResNet-50-R [1]                  |   56.63        |31.14        |25.5    |
> > |RNAS-CL-IL-WRT-50 (Ours) |    61.5         |33.30       |8.5       |
> >
> > Citations
> >
> > [1] Hanxun Huang, Yisen Wang, Sarah M. Erfani, Quanquan Gu, James Bailey, and Xingjun Ma. Exploring architectural ingredients of adversarially robust deep neural networks. In Advances in Neural Information Processing Systems, pp. 5545–5559, 2021.

---

### Official Review · Reviewer_xKFw · 2022-11-01

**Confidence:** 4
**Correctness:** 3
**Technical Novelty And Significance:** 2
**Empirical Novelty And Significance:** 3
**Recommendation:** 6

**Clarity, Quality, Novelty And Reproducibility:**

Clarity is good. The novelty is on the borderline. No code, so reproducibility is doubtful.

**Strength And Weaknesses:**

==== Strength ====
- The first work that applies knowledge distillation-based NAS for adversarial robustness.
- Intuitively, the proposed method is reasonable.
- The paper writing is clear, the method description is easy to understand.

==== Weakness and Questions ====

1. Unsupported statements and missing discussions.

a) This paper claims in the introduction that RNAS-CL is ``the first NAS method that jointly optimizes accuracy, latency, and robustness against adversarial attacks''. However, quite a few studies have already explored this problem, for example, [1][2][3]. Should remove this statement, and discuss / compare with them.

b) The proposed method adopt ideas from existing methods, should discuss more about these existing work, for example, [4].

2. Weakness and questions about the experiments.

a) Missing comparisons with robustness NAS studies like [1][2][3].

b) Necessary baseline results are missing. Table 1 shows the performance of the architectures discovered by different methods on CIFAR-10. The RNAS-CL-* models are searched with the Wide-ResNet teacher model, but the performance of the teacher model is not mentioned. The performance of Wide-ResNet should be given as baseline results. In particular, I wonder if the searched architecture outperforms the Wide-ResNet model, which is very important to verify the effectiveness of the proposed method;

c) Why are only the results of using WRT as the teacher model on CIFAR-10 reported in Table 1? For ImageNet-100, the results with different models as teacher models are provided in the appendix. How will the teacher model's architecture or performance influence the performance of discovered architecture.

d) Some essential experiment settings are missing. For example, what’s the step number for PGD attack in the paper?

[1] Xie, Guoyang, et al. "Tiny adversarial mulit-objective oneshot neural architecture search." arXiv preprint arXiv:2103.00363 (2021).

[2] Ning, Xuefei, et al. "Discovering Robust Convolutional Architecture at Targeted Capacity: A Multi-Shot Approach." arXiv preprint arXiv:2012.11835 (2020).

[3] Yue, Zhixiong, et al. "Effective, efficient and robust neural architecture search." 2022 International Joint Conference on Neural Networks (IJCNN). IEEE, 2022.

[4] Li, Changlin, et al. "Block-wisely supervised neural architecture search with knowledge distillation." Proceedings of the IEEE/CVF Conference on Computer Vision and Pattern Recognition. 2020.

Another minor suggestion: You should use ``'' for quotation marks in LaTex.

**Summary Of The Paper:**

This paper utilizes neural architecture search to discover neural network architectures that are adversarially robust and with low latency at the same time. In particular, they introduce an adversarially robust teacher model for cross-layer distillation in the training and search process to improve the adversarial robustness of the model, and specifically search which layers of the teacher model are used for feature distillation.

**Summary Of The Review:**

My concerns about this paper mainly come from the insufficient experimental verification. I may consider improving the score if the authors could provide the missing experiments and results.

---

> ### Author Response · Authors · 2022-11-12
> **Response to Reviewer xKFw**
>
> We would like to thank the reviewer for his comments on the paper. We address all the points raised by the reviewer below. We hope that the reviewer will reconsider the evaluation in light of our responses.
>
> 1. (a) We have updated the statement. We compared our method against [1][2][3] in Table 1. Compared to [1, 3], RNAS-CL models achieve ~20-30% higher robust accuracy. Compared to [2], RNAS-CL models achieve similar robust accuracy; however, RNAS-CL exceeds [2] by ~3% in terms of clean accuracy.
>
> 1. (b) As suggested by the reviewer, we have updated the related work section with a discussion on [1][2][3][4].
>
>
> 2 (a) We have compared our method against [1][2][3] in Table 1.
>
> (b) We have reported the performance of various teacher models in Table 7. RNAS-CL-WRT models, while being 15X smaller than the teacher model, achieve ~1.5% higher clean accuracy and ~4% lower PGD accuracy.
>
> (c) We conducted experiments using Robust ResNet-50 and ResNet-18 as teacher models in Table 2 of the revised paper. All RNAS-Cl models, while achieving similar clean accuracy, exceed its counterpart by more than 10% in PGD accuracy. RNAS-CL-R50 achieves higher robust accuracy than RNAS-CL-R18 and RNAS-CL-WRT-34. However, ResNet-50 has the lowest PGD accuracy among the teacher models, suggesting that the teacher's architecture influences the student's performance more than the teacher's performance. The higher number of teacher layers allows more options for the student layer to learn from, leading to better robustness.
>
> (d) Throughout the paper, we used a 20-step PGD attack to evaluate all models until mentioned otherwise. We have added this detail to the paper. Link to the code was attached in the abstract.

---

> > ### Comment · Reviewer_xKFw · 2022-11-22
> > **Thanks for the response**
> >
> > Thanks for the added experiments. As this is my major concern. I'll increase the score.
> >
> > Currently, I think the paper's novelty is on the borderline. Nevertheless, If the authors can verify the proposed distillation method in broader NAS applications, the contribution can be enhanced.

---

### Decision · Program_Chairs · 2023-01-20

**Decision:**

Reject

**Justification For Why Not Higher Score:**

Authors simply missed a significant set of works which are highly relevant in their submission. Authors mistakenly claimed to be the first to address robustness and NAS. Although authors fixed the claim and added lots of new experiments, the writing still misses these additional references. Both introduction and related work need a major revision addressing these work.

**Justification For Why Not Lower Score:**

N/A

**Metareview: Summary, Strengths And Weaknesses:**

The paper is proposing a neural architecture search which can satisfy both adversarial robustness and low latency as well as high accuracy. It builds on knowledge distillation methods and also choose which layer to distill. The paper is reviewed by three experts and all agreed on merits and limitations of the paper. The paper is interesting and performs very well. However, it had some issues in terms of related work. It missed some important work which also addressed adversarial robustness and NAS. Moreover, after these works are considered reviewers raised novelty issues. After my personal reading of the paper, I agree with the merits and the weakness of missing related work. However, I do not agree with the novelty issue. I believe the related work issue is still largely unsolved. Authors did provide more results comparing with these method and the results are promising. However, the paper is not revised accordingly. These methods are not mentioned and discussed in detail. Introduction still reads as if these methods do not exist. Hence, the paper needs a major revision in terms of introduction and related work discussing these methods pointed by the reviewers.